# High-throughput screening of prostate cancer risk loci by single nucleotide polymorphisms sequencing

Peng Zhang[1,2], Ji-Han Xia[3], Jing Zhu[2], Ping Gao[3], Yi-Jun Tian[2], Meijun Du[2], Yong-Chen Guo[2], Sufyan Suleman[3], Qin Zhang[3], Manish Kohli[4], Lori S. Tillmans[5], Stephen N. Thibodeau[5], Amy J. French[5], James R. Cerhan[6], Li-Dong Wang[1], Gong-Hong Wei [3] & Liang Wang [2]

Functional characterization of disease-causing variants at risk loci has been a significant challenge. Here we report a high-throughput single-nucleotide polymorphisms sequencing (SNPs-seq) technology to simultaneously screen hundreds to thousands of SNPs for their allele-dependent protein-binding differences. This technology takes advantage of higher retention rate of protein-bound DNA oligos in protein purification column to quantitatively sequence these SNP-containing oligos. We apply this technology to test prostate cancer-risk loci and observe differential allelic protein binding in a significant number of selected SNPs. We also test a unique application of self-transcribing active regulatory region sequencing (STARR-seq) in characterizing allele-dependent transcriptional regulation and provide detailed functional analysis at two risk loci (*RGS17* and *ASCL2*). Together, we introduce a powerful high-throughput pipeline for large-scale screening of functional SNPs at disease risk loci.

[1] Henan Key Laboratory for Esophageal Cancer Research, The First Affiliated Hospital of Zhengzhou University, 40 Daxue Road, 450052 Zhengzhou, Henan, China. [2] Department of Pathology, MCW Cancer Center, Medical College of Wisconsin, 8701 Watertown Plank Road, Milwaukee, WI 53226, USA. [3] Biocenter Oulu, Faculty of Biochemistry and Molecular Medicine, University of Oulu, Aapistie 5 A, 90220 Oulu, Finland. [4] Department of Oncology, Mayo Clinic, 200 First Street SW, Rochester, MN 55905, USA. [5] Department of Laboratory Medicine and Pathology, Mayo Clinic, 200 First Street SW, Rochester, MN 55905, USA. [6] Department of Health Sciences Research, Mayo Clinic, 200 First Street SW, Rochester, MN 55905, USA. These authors contributed equally: Peng Zhang, Ji-Han Xia. Correspondence and requests for materials should be addressed to L.-D.W. (email: ldwang2007@126.com) or to G.-H.W. (email: gonghong.wei@oulu.fi) or to L.W. (email: liwang@mcw.edu)

Since 2005, more than 2000 genome-wide association studies (GWASs) have been published, identifying loci associated with susceptibility to over 1000 unique traits and common diseases[1,2]. With an eventual goal of finding new therapies and preventative measures for these diseases, efforts are now focused on determining the functional underpinnings of these associations. For example, prostate cancer GWASs have been extremely productive, yielding over 100 single-nucleotide polymorphisms (SNPs) with risk association[3,4]. Importantly, a significant number of these have subsequently been validated in well-powered, large case–control studies. Despite exceptional success, however, we are faced with the tremendous challenge of how to interpret these emerging results. There is currently a substantial knowledge gap between SNP–disease associations derived from GWASs and an understanding of how these risk SNPs contribute to the biology underpinning human diseases[5].

As most risk SNPs have been found in non-coding regions of the genome, with many residing some distance from nearby annotated genes, it is believed that many of these (or their closely linked causal SNPs) will be located in regulatory domains of the genome that control gene expression rather than in coding regions that directly affect protein function[3,4]. Due to lack of more effective screening approaches, identification and functional characterization of disease-causal SNPs remains as a significant challenge. For a given GWAS locus, the SNP with the lowest $P$ value is not necessarily causal. Any SNP in linkage disequilibrium (LD) with a reported risk SNP may be causal, and the number of such LD SNPs are often from dozens to thousands[6]. To associate GWAS variants with regulatory elements in the genome, epigenomic profiling such as ChIP-seq, DNase-seq, and their derivatives (ChIP-exo and ChIP-nexus) have been developed[7–11]. Several computational programs have also been developed to integrate epigenomic landscapes with GWAS SNPs[12–16]. These profiling analyses and computational programs have been widely used and help facilitate discovery of candidate regulatory SNPs.

However, there remains a significant challenge from the knowledge-based prediction to functional validation. Currently, to experimentally validate putative SNPs for regulatory potential, the commonly used assays include electrophoretic mobility shift assays (EMSA) and reporter assays. EMSA can test whether a given SNP influences binding ability of a transcription factor (TF) to the regulatory element while a gene reporter assay can test the effect of a SNP on promoter or enhancer activity. More recently, CRISPR/Cas9-based gene editing technology has emerged as important tool to evaluate the effect of a single-nucleotide variant[17,18]. Although these current methods have enabled functional characterization of regulatory variants at some GWAS loci, the progress is extremely slow. Given the huge number of disease-associated regulatory variants, high-throughput methods are urgently needed to overcome limitations of these one-assay-one-SNP approaches.

One existing high-throughput method is to calculate allele-specific read counts from available ChIP-seq data. Significant deviation of read counts between two alleles indicates allelic binding preference for the unique TF. However, to be informative, the SNPs of interest need to be heterozygous in the tested cell line. For a large group of SNPs, it is difficult to find cell lines with heterozygous status in all (or most) candidate SNPs. In this study, we report a new massively parallel sequencing technology to distinguish potentially functional from nonfunctional SNPs. We apply the sequencing technology (named as single-nucleotide polymorphisms sequencing or SNPs-seq) to examine potential functional SNPs at prostate cancer-risk loci. We also test a unique application of self-transcribing active regulatory region sequencing (STARR-seq)[19–21] in characterizing allele-dependent transcriptional regulation and provide detailed functional analysis at

two risk loci (*RGS17* and *ASCL2*). This study introduces a powerful experimental pipeline to functionally screen for regulatory SNPs at GWAS-defined risk loci.

## Results

**Principles of functional SNP sequencing**. To precisely determine allelic protein-binding differences with high-throughput capacity, we developed the massively parallel sequencing technology based on the principal that protein-bound DNA oligos will be retained in a protein purification column after extensive washes to remove free oligos. Following recovery from the column, SNP-containing oligos that are bound to nuclear proteins can be sequenced to determine allele-dependent protein binding (Fig. 1). Because this technology will test the SNP-dependent protein-binding difference, we defined the sequencing technology as SNPs-seq. Meanwhile, to identify DNA sequences that act as transcriptional enhancers in a direct, quantitative, and genome-wide manner, the STARR-seq has been developed by taking advantage of the knowledge that enhancers can work independent of their relative locations[19–21]. This assay inserts candidate DNA sequences downstream of a super core promoter and allows the active enhancers to transcribe themselves. Such a direct coupling of candidate sequences to enhancer activity enables the parallel evaluation of millions of DNA sequences from arbitrary sources. We hypothesize that STARR-seq can quantitatively detect the enhancer activity of allele-specific sequences by measuring the RNA abundance among cellular RNAs. The principle and workflow of STARR-seq are shown in Supplementary Fig. 1 and 2.

**Study design**. We first examined our prostate-specific expression quantitative trait loci (eQTL) dataset[22] and functional annotation databases to determine candidate functional SNPs at prostate cancer GWAS loci. To further screen for candidate functional SNPs, we then performed SNPs-seq to determine allele-dependent protein-binding differences at these SNP sites simultaneously. To evaluate if these SNPs also showed differential transcriptional regulation, we performed STARR-seq and determined allelic read count of SNP-containing RNA transcripts. Finally, we applied a series of functional assays to validate the sequencing results. The overall study design is depicted in Fig. 2.

**Selected candidate SNPs**. We previously performed eQTL analysis at 100 prostate cancer-risk intervals covering 146 risk SNPs in 471 normal prostate tissues and identified 51 loci showing significant *cis*-eQTL signals ($P$ value threshold of 1.96E−07), which were involved in a total of 2208 SNPs and 88 individual genes[22]. To select candidate functional SNPs from this reported eQTL SNPs, we examined prostate-specific ChIP-seq data[23] and searched HaploReg database[14]. These analyses identified 255 potential functional SNPs involving 35 genes with eQTL $P \le 1.96E−07$. To expand the candidate SNP list, we selected an additional 51 SNPs involving 10 genes with eQTL $P$ value between 0.05 and 1.96E−07. Finally, we selected 68 SNPs that were either reported risk SNPs or in LD with these SNPs but did not show any association with any reference genes in ±1 Mb regions. Overall, we selected 374 SNPs at 33 separate risk loci including 755 unique sequences (369 SNPs with one variant, 3 SNPs with 2 variants, and 2 SNPs with 3 variants). Most of the candidate SNPs were located at introns or intergenic regions (Supplementary Data 1 for SNP names and chromosome coordinates, corresponding genes, eQTL $P$ values and ChIP-seq score).

**Quality check of SNPs-seq libraries**. To determine allele-dependent protein binding, we mixed 755 unique ds-oligos equally (5.05 nM each oligo) and used 264 ng of the oligo pool

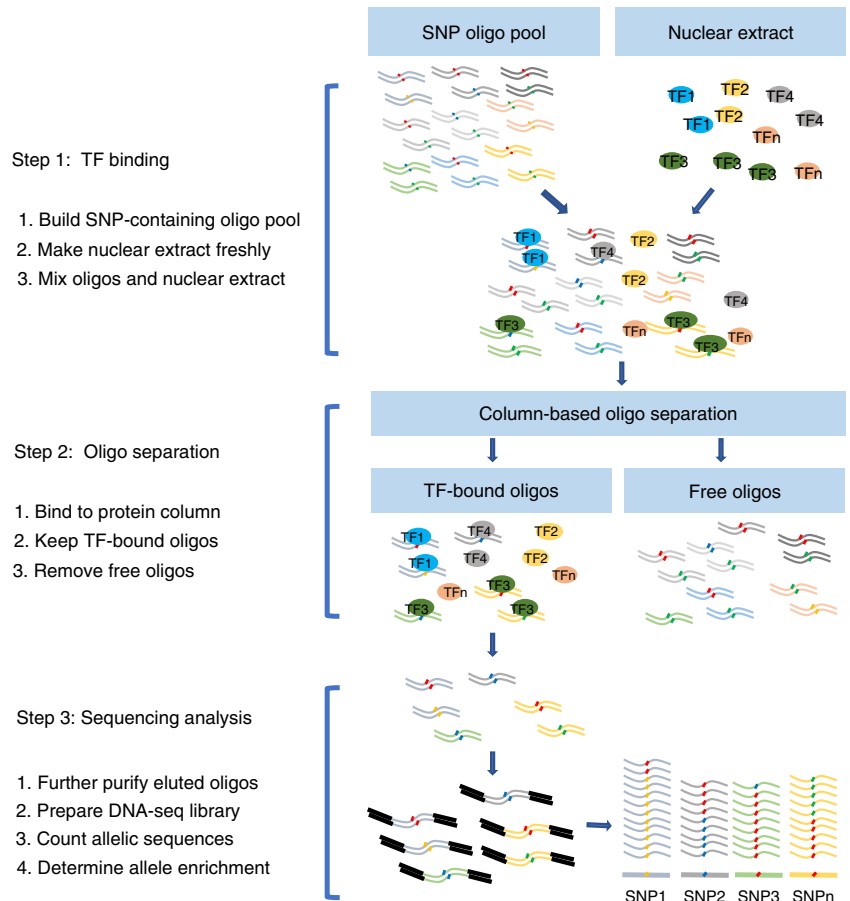

**Fig. 1** Workflow of SNPs-seq. The SNPs-seq includes three key steps: binding of SNP-containing oligos to nuclear protein, separation of protein-bound oligos from protein-free oligos, and sequencing library preparation and analysis

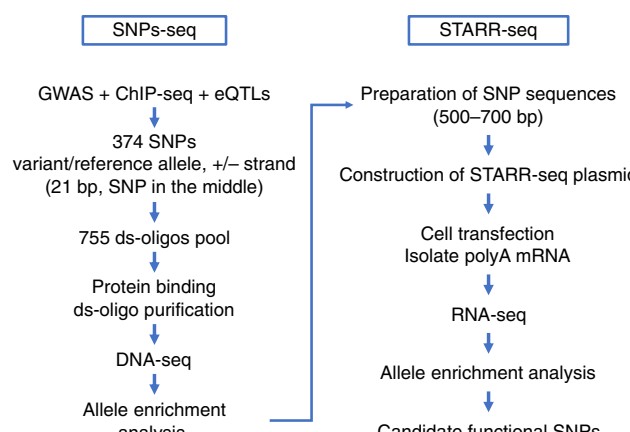

**Fig. 2** Overall study design. SNPs-seq workflow (left panel): 374 SNPs were selected from analyzing GWAS, ChIP-seq, and eQTL data. SNP-containing oligos were synthesized, followed by annealing the positive and negative strands to make ds-oligos. By mixing ds-oligos with nuclear extract, the protein-bound ds-oligos were separated and used for library preparation and allele-specific sequencing analysis. STARR-seq workflow (right panel): Significant SNPs selected from SNPs-seq were PCR-amplified, pooled and inserted into STARR-seq vector. After co-transfecting LNCaP cells, the mRNAs from transfected cells were isolated and used to make STARR-seq library for allele-specific sequencing analysis

(25.23 fmols each oligo) for protein-binding assay. After extensive washes, the average yield of the eluted protein-bound oligos was 6–10 ng, accounting for 2–4% of original input (Supplementary Fig. 3a). Due to the addition of adaptor sequences during library preparation, we expected to see the library size at ~161 bp (21 bp oligo + 140 bp adaptors). For the libraries prepared from input controls, the size distribution was as expected (sharp band at ~161 bp). Because nuclear extract may contain low level "naked" fragmented DNA, library sizes from nuclear extract-bound oligos varied from ~150 to ~500 bp (Fig. 3a).

Owing to single-nucleotide differences between variant and reference alleles, we only counted sequence reads with a perfect match to one of 755 unique oligo sequences. Depending on the individual library, the SNP-specific sequences accounted for ~70% of raw reads, ranging from 57% in protein-bound ds-oligos to 94% in input control ds-oligos (Fig. 3b). Low on-target reads in the protein-bound ds-oligos were expected since some fragmented DNAs from nuclear extract were also sequenced. To examine reproducibility of the technology, we performed each assay with a technical replicate and tested two different protein isolation platforms (Affymetrix and Signosis). Correlation coefficient analysis showed high reproducibility in all technical replicate pairs ($R^2 > 0.95$) (Fig. 3c) and between different platforms ($R^2 > 0.86$) (Supplementary Fig. 3b). We also estimated the effect of input oligo quantity on read counts by comparing 88, 264, and 792 ng ds-oligo pool as input. This analysis showed no clear read count difference among the three inputs (Supplementary Data 2).

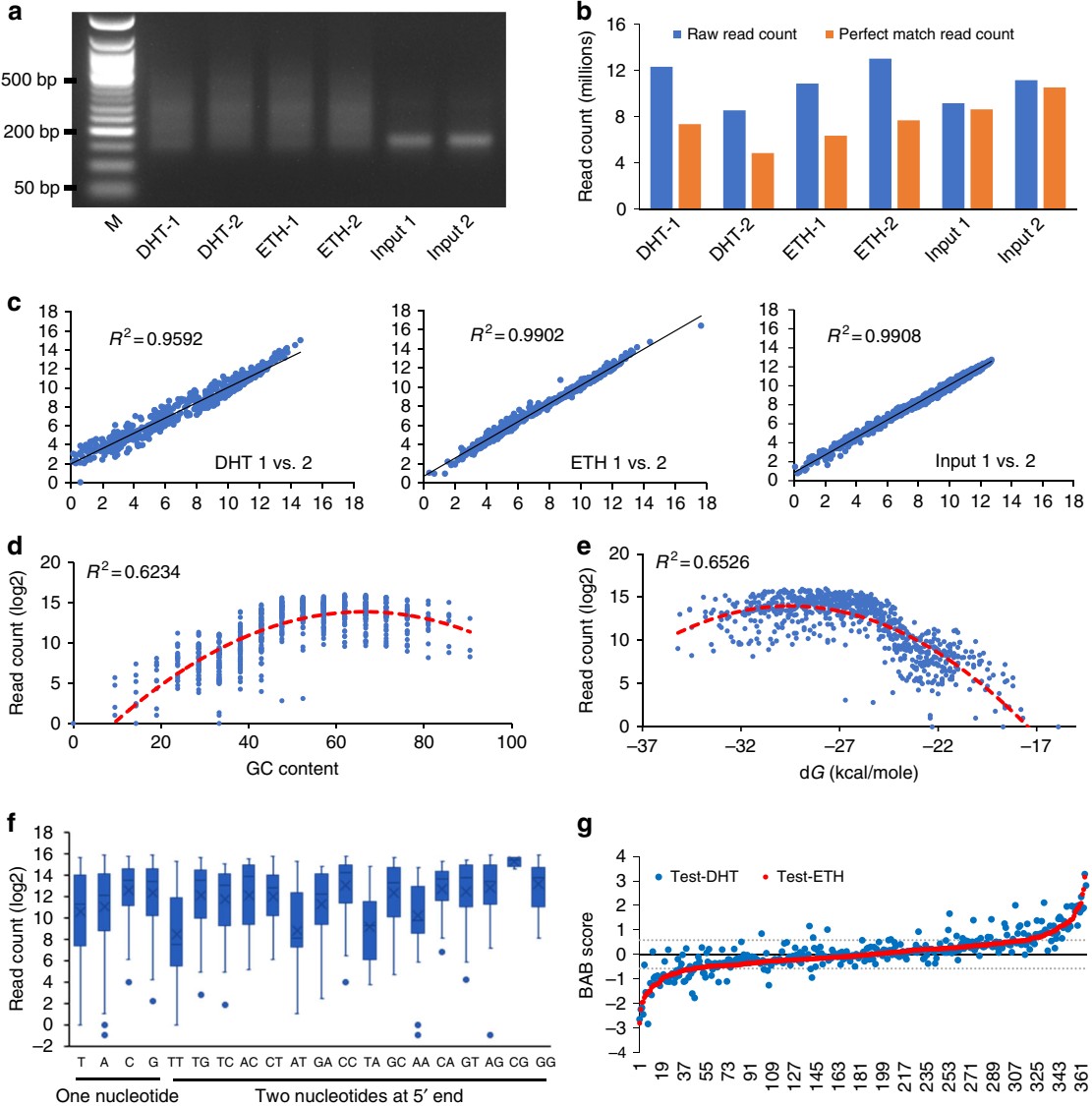

**Fig. 3** SNPs-seq data analysis and BAB score distribution. **a** Size distribution of SNPs-seq libraries. Test samples show 150–500 bp length while input samples show ~160 bp. **b** Mapping of allele-specific read counts. Percentage of mapped read count is ~60% for test samples and ~95% for input controls. **c** Correlation between technical replicates. Mapped read counts were first transformed to log2 values, and then plotted along $x$ (replicate 1) and $y$ (replicate 2) axis. **d** Association of read count with GC content. Low GC content is significantly associated with low read count. **e** Association of read counts with delta $G$ value. More thermostability of oligo duplex (lower delta $G$) contributes to higher read counts in SNPs-seq library. **f** Association of read counts with nucleotides at 5′ end. The nucleotides A, T, AA, AT, TA, or TT at 5′ end have the lowest read counts among all tested oligos. The upper, middle, and lower bounds of boxes represent the 75th, 50th, and 25th percentile of the values, respectively. The whiskers represent 95th to 5th percentile. **g** Overall distribution of BAB scores among the 374 tested candidate SNPs. The red line represents BAB scores in ETH group while the blue dots represent the corresponding BAB scores from the DHT group

**Effect of oligo content on read counts**. Although we attempted to pool these oligos equally, we observed a significant read count difference between different SNPs. Median read count per allele was 10,210, ranging from 0 to 1,294,410 in 20 different SNPs-seq assays including 6 assays with Affymetrix kit and 14 assays with Signosis kit. The significant variations of read counts among these oligos could be caused by multiple factors including poor oligo quality, inaccurate quantity of input, nucleotide composition, and thermostability of each oligo. We first evaluated the effect of GC content and thermodynamic nature on the read counts. This analysis revealed significant correlation of read counts with oligo GC content ($R^2 = 0.6234$), with low GC content toward low read counts. Among 755 unique oligo sequences, 49 sequences had low GC content (25% GC as cutoff) and 50% of these low GC oligos

had low read counts (read counts 50 as cutoff) (Fig. 3d). Accordingly, the read counts were also associated with Tm values ($R^2 = 0.6458$, Supplementary Fig. 3c). We also calculated delta G using online program (http://primerdigital.com/tools/)[24] to determine thermostability of each oligo and observed a significant association ($R^2 = 0.6526$, Fig. 3e). Clearly, higher thermostability of these single-stranded oligos increased annealing efficiency to form double-stranded oligos, hence providing more templates for sequencing library preparation.

We then tested the effect of nucleotide compositions at 5′/3′ ends on read counts. We grouped the 755 oligos into different groups, based on their nucleotide compositions. When compared to G/C, single-nucleotide A/T at either 5′ or 3′ end significantly reduced read counts of corresponding oligos. When comparing

16 possible two nucleotide combinations at either 5′ or 3′ ends, the combinations of AA, AT, TA, and TT showed the lowest read counts (Fig. 3f, Supplementary Fig. 3d). Clearly, overall GC content (GC%, Tm, and thermostability) and 5′/3′ end nucleotides have a significant effect on read count in the final sequencing libraries.

**SNPs with allele-dependent protein binding**. To evaluate allelic protein-binding differences, we calculated biased allelic binding (BAB, see Methods for detail) scores for all 374 candidate SNP sites. Example of the score calculation for SNP rs7123418 is illustrated in Supplementary Fig. 4 and the distribution of BAB scores is shown in Fig. 3g. When comparing variant allele to reference allele, protein-binding capacity varied significantly, ranging from ~14.37-fold decrease (BAB score = −3.85) to ~15.20-fold increase (BAB = 3.93). When applying an absolute BAB score of ≥ 0.58 (1.5-fold difference between variant and reference alleles) and eQTL $P$ value ≤ 1.00E−05 as significant cutoffs, 101 of the 374 candidate SNPs met the selection criteria (Supplementary Data 3).

**Quality check of STARR-seq libraries**. To test whether the candidate SNPs selected from SNPs-seq also showed allele-dependent enhancer activities, we constructed a plasmid library by inserting PCR-generated fragment pool (see Supplementary Data 3 for primer sequences) into STARR-seq vector. After transfecting LNCaP cells with the plasmid library, we isolated polyA + mRNA from the cells and performed RT-PCR to amplify the target sequences. As expected, the RT-PCR product showed a size of ~600 bp (Supplementary Fig. 5a). To effectively sequence through SNP sites, we sheared the PCR products into 100–150 bp before preparing final sequencing library. We mapped the sequence reads to 96 amplicons covering all 101 selected candidate SNPs. The mapped sequences accounted for an average of 76% raw reads, ranging from 70% in PolyA + mRNA group to 81% in plasmid DNA control group (Supplementary Fig. 5b). Pairwise comparison between technical replicates showed significant correlations in each of three technical repeat pairs ($R^2 \geq$ 0.95) (Supplementary Fig. 5c).

**SNPs with allele-dependent enhancer activities**. For the 202 alleles (101 SNPs), the median read depth per allele was 20,961, ranging from 0 to 622,402. To select sequences with regulatory potential, we first transformed the read counts to RPM (read count per million mapped sequences) and then compared allele-specific read counts between test samples and input control samples. This analysis identified 56 SNPs with at least one allele showing 1.5-fold difference between test samples and input controls. To determine allele-dependent enhancer activity of the 56 SNPs, we calculated biased allelic enhancer (BAE, see Methods for detail) score using allele-specific read counts and observed 20 SNPs with absolute BAE score ≥ 0.58. The overall distribution of the BAE score is shown in Supplementary Fig. 5d (Supplementary Data 4 for STARR-seq read counts and BAE scores for all 101 SNPs). These SNPs were associated with 11 separate eQTL genes, including *LOC284581* (1 SNP), *NOL10* (1 SNP), *RAB17* (4 SNPs), *RGS17* (1 SNP), *HCG4B* (1 SNP), *PCAT1* (3 SNPs), *CTBP2* (2 SNPs), *NCOA4* (1 SNPs), *ASCL2* (4 SNPs), *C14orf39* (1 SNP), and *FAM57A* (1 SNP).

**Functional characterization of selected SNPs**. Based on the BAE score, eQTL $P$ value and ChIP-seq evidence, we selected one SNP (rs13215402) in the *RGS17* region, and three SNPs (rs6579003, rs7123299, and rs7123418) in *ASCL2* region for further analysis. The SNP rs13215402 is located at 4.84 kb upstream of *RGS17* transcription start site. ChIP-seq analysis shows that rs13215402 is

at the center of multiple TF-binding sites including AR, FOXA1, and HOXB13. The SNP site is enriched with active enhancer epigenetic marks, H3K4me1/2 and H3K27ac (Fig. 4a). However, there is no enrichment of silent epigenetic mark H3K27me3 and condensed chromatin regulator EZH2. To further test the TF enrichment, we carried out ChIP assays, followed by quantitative PCR and confirmed the chromatin binding of prostate cancer master regulators AR, FOXA1, and HOXB13 at the SNP site (Supplementary Fig. 6a, b). To assess whether rs13215402 showed allele-specific effect in vivo, we performed ChIP-based allele-specific quantitative PCR (ChIP-AS-qPCR) in the LNCaP cells that are heterozygous for rs13215402. We observed a marked decrease of FOXA1 binding at the variant allele A when compared to the reference allele G (Fig. 4b). To test the regulatory potential of this SNP, we further performed an allele-specific luciferase reporter assay and observed significantly lower signal in variant allele A than reference allele G ($P = 1.13E−05$ in dihydrotestosterone (DHT)-treated cell line and $P = 6.78E−03$ in cell line without DHT treatment) (Fig. 4c). The luciferase activity difference between allele A and allele G was even more significant ($P ≤ 3.09E−09$) when replacing pGL4.28 minimal promoter with *RGS17* promoter regardless DHT treatment (Fig. 4d). This result is consistent with STARR-seq showing significantly lower read counts in variant allele A than reference allele G. The BAE scores were −0.88 in DHT-treated group and −0.52 in cell line without DHT treatment (Supplementary Data 4).

To further delineate regulatory role of the rs13215402, we applied CRISPR interference (CRISPRi) technology to evaluate the repression effect by interfering the SNP region on the RGS17 expression. We designed small guide RNAs to target either rs13215402 A or rs13215402 G allele in LNCaP cell line (heterozygous G/A for rs13215402). We then quantified the expression of *RGS17* and observed its downregulation by either allele G or allele A interference. Compared to non-target control (NTC), the downregulation of *RGS17* expression was statistically significant in allele G ($P = 0.028$) but not allele A ($P = 0.633$) (Fig. 4e). We also applied CRISPR/Cas9-based genome editing technology in the 22Rv1 cell line (heterozygous G/A for rs13215402) with an aim of creating subclones with homozygous alleles (A/A or G/G) and determining direct effect of these different genotypes on *RGS17* expression. We successfully generated two subclones with genotype G/G (Supplementary Fig. 7a) but did not receive subclones with genotype A/A. The quantitative RT-PCR analysis showed 3-fold increase of *RGS17* expression in subclones with homozygous G/G ($P ≤ 3.80E−04$) when compared to parental cell line with heterozygous G/A (Fig. 4f). Correspondingly, eQTL analysis using 467 benign prostate tissues showed an association of the variant allele A with reduced expression of *RGS17* (eQTL $P =$ 9.61E−31) (Fig. 4g). Additionally, eQTL analysis using three other independent datasets encompassing 602 prostate tumor samples[23] also showed the significant reduction of the gene expression in patients with allele A (Supplementary Fig. 7b).

The haplotype SNPs (rs6579003, rs7123299, and rs7123418) are located at ~59.8 kb upstream of *ASCL2* transcription start site. ChIP-seq data show various degrees of overlapping with TF-binding sites including AR, FOXA1, and HOXB13 (Fig. 5a). Because they are clustered in a small 68 bp region, the three SNPs were tested as haplotypes including two common (C–G–C and A–A–A) and one rare haplotype (A–G–C). To test whether the haplotype SNPs also demonstrated allele-specific protein binding in vivo, we performed ChIP-AS-qPCR assay in VCaP cell line with heterozygous genotype for rs7123299 and observed an increased chromatin binding of FOXA1 and HOXB13 at the allele A compared to the allele G (Fig. 5b). Interestingly, the binding ability of the two alleles was switched upon DHT treatment in VCaP cells, showing reduced recruitment of AR, FOXA1, and

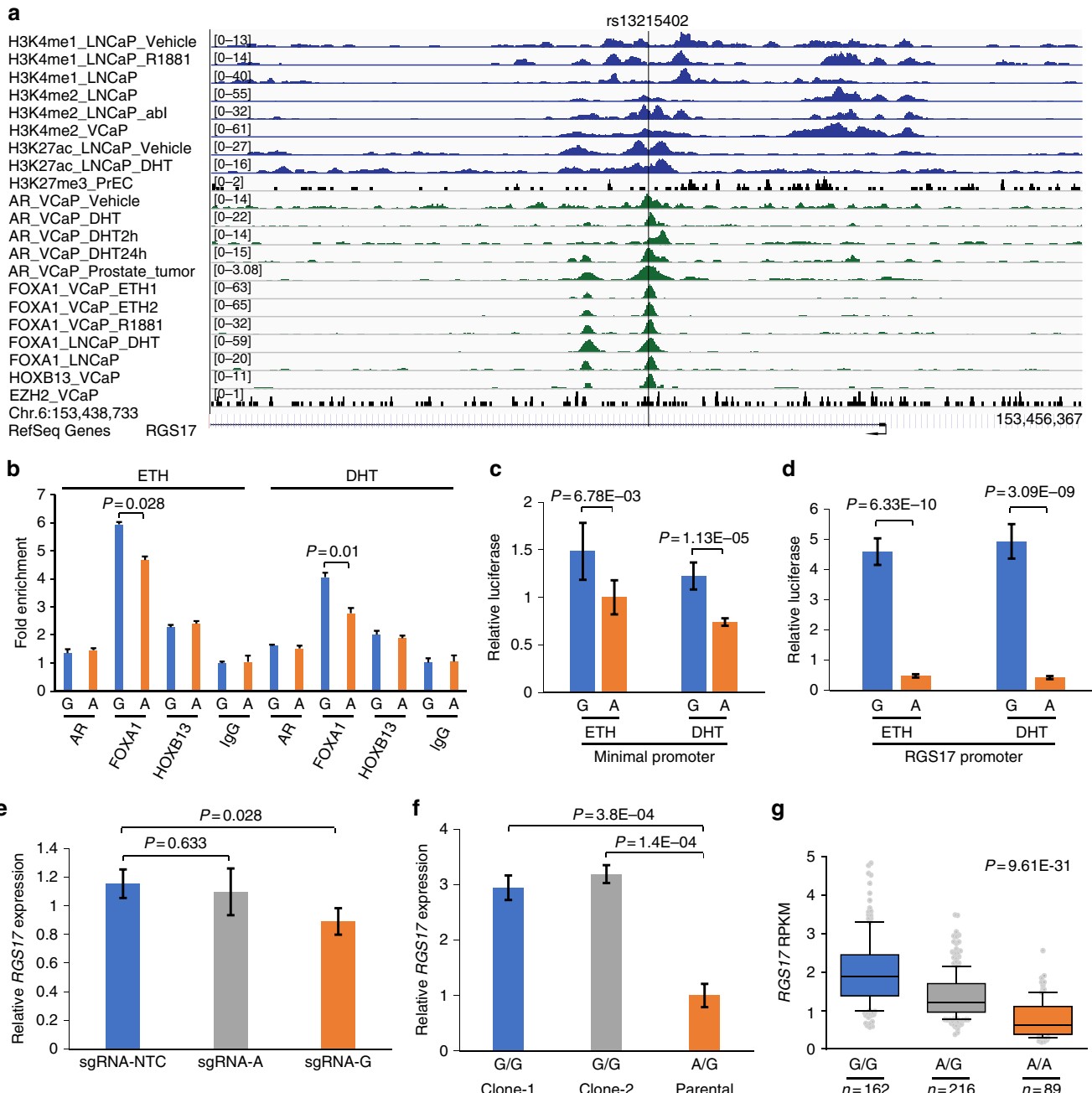

**Fig. 4** Allele-dependent transcriptional regulation at rs13215402. **a** TF enrichment at SNP rs13215402 site by ChIP-seq analysis. The SNP region is occupied by multiple TFs including AR, FOXA1, HOXB13, and active enhancer epigenetic marks including H3K4me1/2 and H3K27ac. **b** TF enrichment at SNP rs13215402 site by ChIP-AS-qPCR analysis. Allele A of this SNP has lower FOXA1 occupancy than allele G in LNCaP cells. The $P$ values were calculated using the two-tailed Student's $t$-test, mean ± s.d. **c**, **d** Luciferase reporter assay at rs13215402 site. The relative luciferase activity for allele A is lower than allele G both under ETH and DHT treatment in LNCaP cells. The $P$ values were calculated using the two-tailed Student's $t$-test, mean ± s.d. **e** Suppression of *RGS17* expression through allele-specific CRISPRi assay. Compared to non-target control (NTC), interference of either allele A or G downregulated *RGS17* expression, with allele G showing statistical significance. The $P$ values were calculated using the two-tailed Student's $t$-test, mean ± s.d. **f** Elevated *RGS17* expression after converting genotype of rs13215402 G/A to G/G in 22Rv1 cells by CRISPR/Cas9. Compared to parental cell line 22Rv1 with G/A genotype, two subclones of 22Rv1 with G/G genotype show threefold increase of *RGS17* expression. The $P$ values were calculated using the two-tailed Student's $t$-test, mean ± s.d. **g** eQTL analysis between rs13215402 and *RGS17*. Compared to G/G genotype, the A/A genotype is associated with lower expression of *RGS17* in benign prostate tissues[22]. The upper, middle, and lower bounds of boxes represent the 75th, 50th, and 25th percentile of the values, respectively. The whiskers represent 95th to 5th percentile. The $P$ values were calculated using the correlation/trend test (genotype association test in Golden Helix)

HOXB13 at the rs7123299 allele A. We also performed a luciferase reporter assay to examine enhancer activity of different haplotypes in vitro and found much higher regulatory activity of the DNA fragment carrying variant haplotype (A–A–A) than the other two haplotypes (C–G–C and A–G–C) ($P < 0.002$, Fig. 5c).

To see whether the haplotype-based reporter assays were consistent with STARR-seq, we re-examined STARR-seq data to construct the haplotypes by counting sequence reads that covered all three SNPs. Among 96 amplicon fragments tested, three were found to span across these three SNPs. We counted each

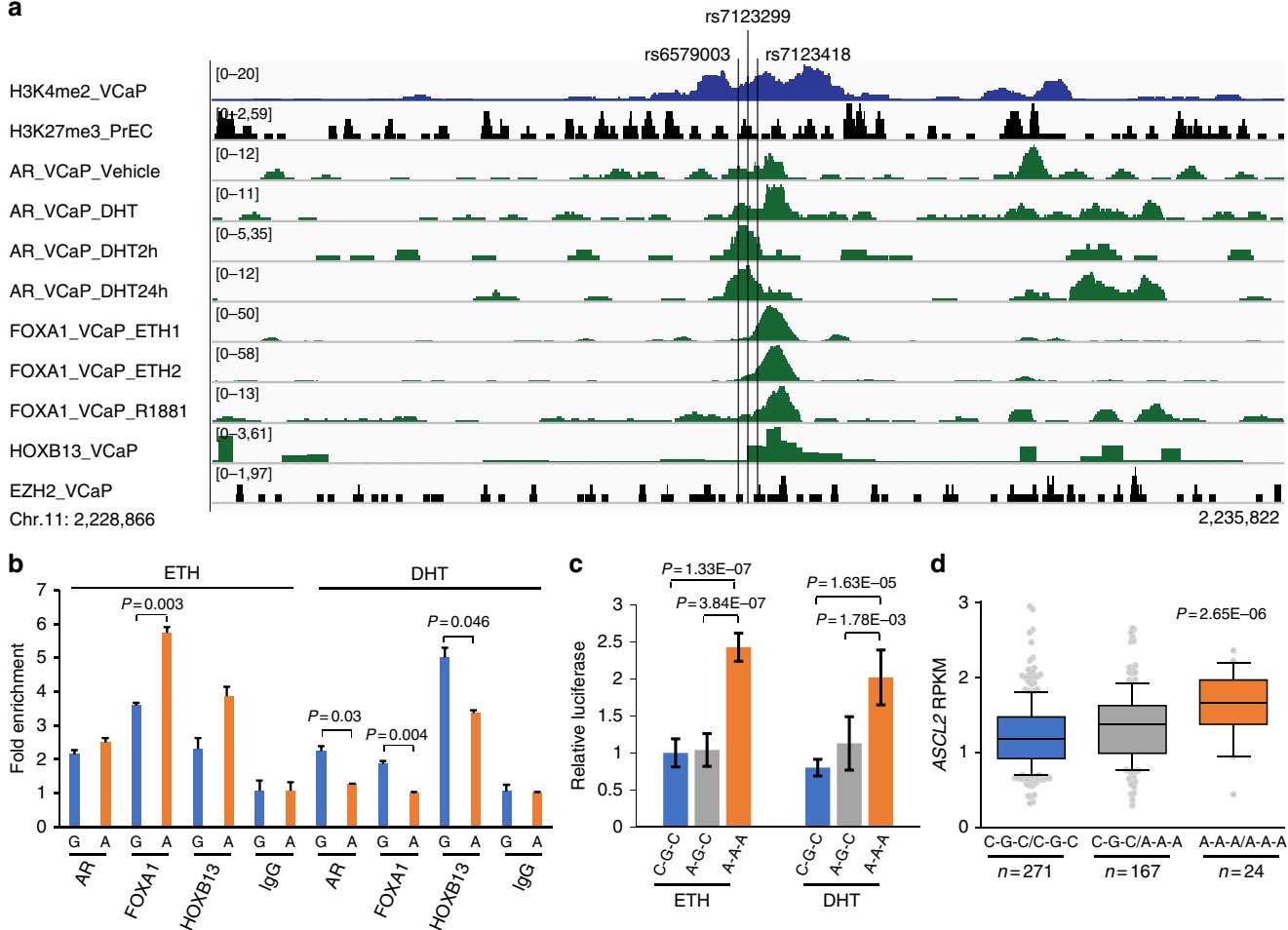

**Fig. 5** Allele-dependent transcriptional regulation at a haplotype region containing rs6579003, rs7123299, and rs7123418. **a** TF enrichment at SNPs (rs6579003, rs7123299, and rs7123418) by ChIP-seq analysis. The region is occupied with TFs such as AR, FOXA1, and HOXB13, and enriched with active chromatin mark H3K4me2. **b** TF enrichment at SNP rs7123299 site by ChIP-AS-qPCR analysis. The rs7123299 allele A shows higher binding ability to FOXA1 and HOXB13 than allele G in VCaP cells under ETH treatment but effect of the two alleles are switched under DHT treatment. The *P* values were calculated using the two-tailed Student's *t*-test, mean ± s.d. **c** Luciferase reporter assay at three haplotype SNPs (rs6579003, rs7123299, and rs7123418) in LNCaP cells. The relative luciferase activity for haplotype A–A–A is higher than haplotype C–G–C in both ETH and DHT treatment. The *P* values were calculated using the two-tailed Student's *t*-test, mean ± s.d. **d** eQTL analysis between haplotype-based genotypes (rs6579003, rs7123299, and rs7123418) and *ASCL2*. Compared to haplotype genotype CGC/CGC, the AAA/AAA confers higher expression of *ASCL2* in benign prostate tissues. The upper, middle, and lower bounds of boxes represent the 75th, 50th, and 25th percentile of the values, respectively. The whiskers represent 95th to 5th percentile. The *P* values were calculated using the correlation/trend test (genotype association test in Golden Helix)

individual haplotype in the three amplicons and calculated the haplotype-based BAE scores (A–A–A as variant and C–G–C as reference). This analysis showed significant enrichment of haplotype A–A–A with an average BAE score = 3.13 (range: 2.98–3.37) in DHT-treated group, and 2.46 (range: 1.91–2.99) in non-DHT group. Clearly, A–A–A haplotype showed an increased enhancer activity when compared to C–G–C haplotype. To further confirm the effect of the haplotype on gene expression, we examined the haplotype-based eQTL in a collection of 462 benign prostate tissues and observed haplotype-dependent dosage effect on *ASCL2* expression with two copies of A–A–A showing the highest gene expression ($P = 2.65\text{E}{-}06$) (Fig. 5d). The cancer tissue-based eQTL analysis in three independent datasets also showed the increased expression of *ASCL2* in patients with haplotype A–A–A (Supplementary Fig. 7c).

**RGS17 as oncogene of prostate cancer.** Previous GWAS has shown that variant allele G of the risk SNP rs1933488 was associated with decreased prostate cancer risk (OR = 0.89)[25].

Because the allele G is in complete LD with variant allele A of the functional SNP rs13215402 reported in this study, we reasoned that the allele A would also be associated with reduced prostate cancer risk. Based on our prostate tissue eQTL analysis, allele A significantly downregulated target gene *RGS17*, consistent with the protective effect of GWAS risk SNP on prostate cancer. To test whether downregulation of *RGS17* inhibited prostate cancer cell growth, we performed cell proliferation assays in the prostate cancer cells using siRNA against *RGS17* and observed greatly reduced cell growth and viability when compared to the cells with control siRNAs (Fig. 6a, b). Additionally, we examined genome-wide CRISPR/Cas9-based loss-of-function screen data for the identification of genes that are essential for cell growth and survival[26], and found critical role of *RGS17* for survival of prostate cancer cells LNCaP and PC3 (Supplementary Fig. 8a, b). Furthermore, by querying prostate cancer datasets, we found that *RGS17* was up-regulated in prostate adenocarcinoma (Fig. 6c). Higher mRNA level of *RGS17* was associated with higher clinical stage and poor

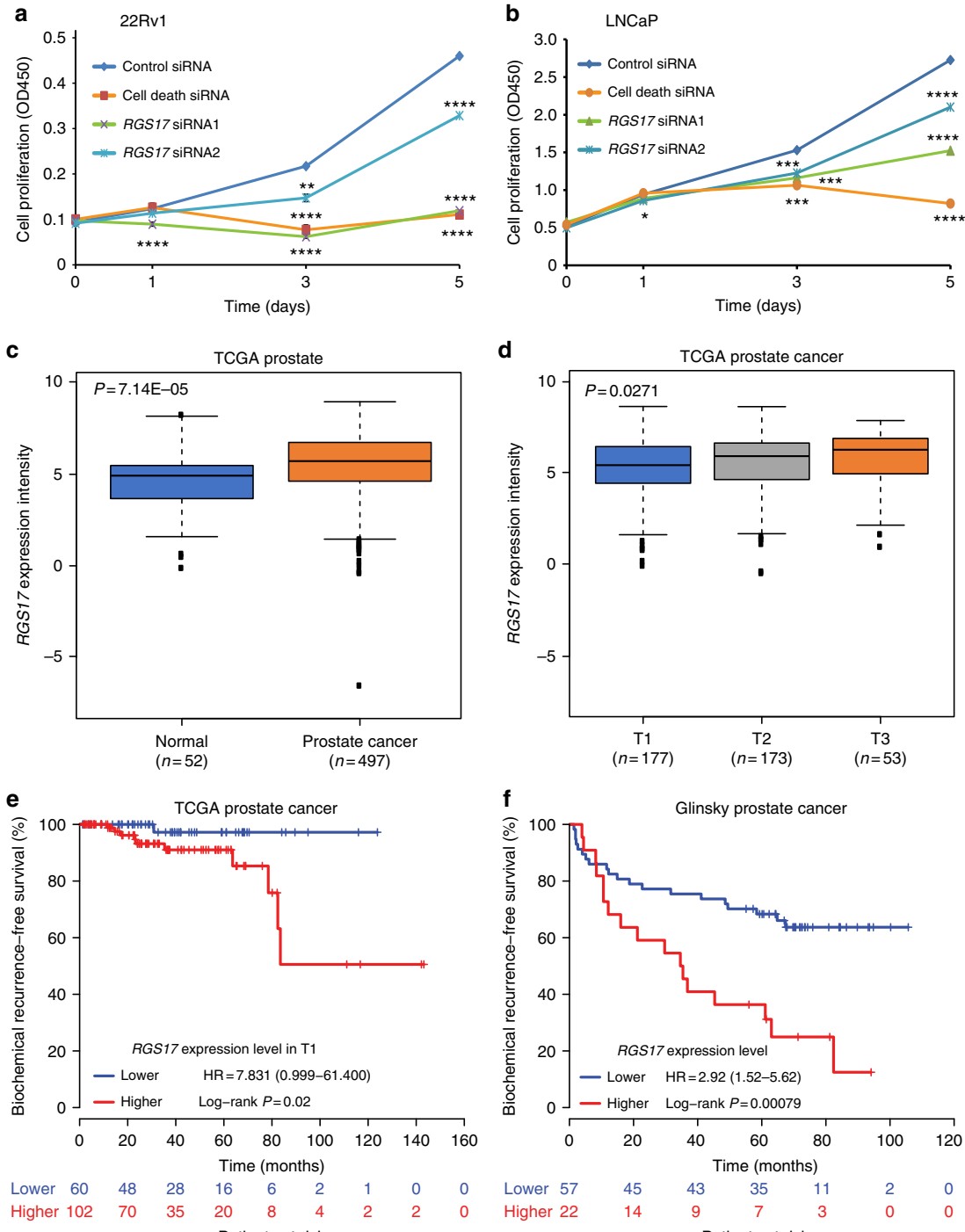

**Fig. 6** Association of the rs13215402-associated gene *RGS17* with cell growth and clinical features for prostate cancer. **a**, **b** Knockdown of *RGS17* by siRNA significantly reduces prostate cancer cell proliferation in 22RV1 and LNCaP cells, measured by XTT colorimetric assay (absorbance at 450 nm, OD450); mean ± s.d. of three independent measurements. **P < 0.01, ***P < 0.001, ****P < 0.0001, two-tailed Student's *t*-test. **c** Expression of *RGS17* is higher in prostate adenocarcinoma than in normal prostate gland in TCGA data. **d** The *RGS17* mRNA expression in patients with T3 stage are higher than T2/T1 stage in TCGA prostate cancer data. In **c–d**, *RGS17* expression intensity is shown as log2 transformed expression value (read per million or RPM by RNA-seq). The upper, middle, and lower bounds of boxes represent the 75th, 50th, and 25th percentile of the values, respectively. The whiskers represent 95th to 5th percentile. Mann–Whitney *U*-tests (**c**) or Kruskal–Wallis test (**d**) were used to evaluate the statistical significance. **e**, **f** The patients with higher mRNA level of *RGS17* have worse recurrence-free survival rate based on TCGA and Glinsky prostate cancer data[52]

prognosis (Fig. 6d–f). Strikingly, *RGS17* expression showed the highest level in prostate cancer among tens of different cancer types in over 10,000 tumor samples (Supplementary Fig. 9a, b), further supporting that *RGS17* is a plausible prostate cancer susceptibility gene.

## Discussion
One of major challenges in the post-GWAS era is the lack of high-throughput assays to screen a large number of candidate SNPs for their potential functional consequences[3,4]. To address this challenge, we report here, for the first time, the development

of the SNPs-seq technology, a high-throughput approach based on massively parallel sequencing strategy to examine SNPs for their protein-binding differences. By applying the technology in prostate cancer-risk regions, we successfully identified over a hundred of SNPs showing allele-dependent protein binding. To evaluate the regulatory potential of these selected SNPs, we further tested the SNP-containing fragments using STARR-seq, a high-throughput enhancer assay, which further defined 20 SNPs with differential transcriptional regulation. Sequential use of the two sequencing technologies may enable comprehensive survey of regulatory SNPs to better understand biological roles of GWAS signals and their functional consequences.

Currently, to determine whether a given SNP is functional, a common approach is to map the SNP to a regulatory element defined by ENCODE and other epigenomic projects[7–9]. Although useful for mapping regulatory genomic regions, none of these datasets provide direct access to allelic binding preferences. One potential solution is to extract read counts at SNP sites from these sequencing data. To date, ENCODE project has collected over six thousands of epigenomic profiling data including ChIP-seq and DNase-seq. Systematic examination of these sequencing data may reveal allele-specific transcriptional control at each SNP site. However, for a specific disease type, there may be no or few epigenomic profiling data available. Importantly, most current epigenomic profiling datasets are based on a handful of cell lines. If a cell line used is not heterozygous for a specific SNP, it will not generate any allelic information. For example, only 96 (25.67%) of 374 candidate SNPs selected in this study are heterozygous in LNCaP cell line. Furthermore, most current epigenomic profiling datasets have relative low coverage at a certain SNP sites, which makes determination of allele-specific binding difficult.

Additionally, allele-specific EMSA and luciferase reporter assays are commonly used in laboratory validation of candidate regulatory SNPs[27–32]. However, both EMSA and reporter assay are not practical if a large number of candidate SNPs are tested. To overcome this limitation, we developed the SNPs-seq by taking advantage of higher retention rate of protein-bound DNAs and high capacity of sequencing technology. This technology applies easy-to-use protein purification columns and determines allele-dependent protein–DNA binding by counting sequence reads. Due to high-throughput nature, this sequencing technology can examine hundreds or even thousands of protein-bound DNA oligos simultaneously, significantly increasing screening capacity for candidate functional SNPs. It is worth mentioning that regulation of gene expression is cell and tissue type-specific, so is allelic binding preference detected by SNPs-seq. When applying the SNPs-seq to detect allele-specific binding difference, cell/tissue type and surrounding environments that are involved in the diseases/phenotypes of interest should be taken into consideration.

Although several high-throughput technologies including MPRA[33–35], MPFD[36], CRE-seq[37–39], STARR-seq[19–21], TRIP[40], FIREWACh[41], and SIF-seq[42] have been developed to evaluate regulatory potential of target sequences, these technologies were initially designed to determine transcriptional regulation at selected genomic regions. So far, no report has been published to test their applications in allele-dependent transcription controls. Due to its simplicity, STARR-seq was selected to examine this unique application. Our result strongly suggests that STARR-seq is capable of characterizing functional SNPs in disease risk regions. Furthermore, we performed in depth evaluation on the regulatory role of two prostate cancer-risk loci and provided strong evidence showing essential role of *RGS17* for the maintenance of the proliferative potential of tumor cells[43,44]. Further characterization of the gene and its regulatory variants will have important implications for developing potential screening strategies to assess prostate cancer predisposition.

Main advantage of SNPs-seq is its high capacity to screen a large number of candidate SNPs in a flexible, easy-to-use and low-cost procedure (Supplementary Data 5 for time and cost estimation). When combining with another high-throughput technology STARR-seq, we may not only detect the allele-dependent protein-binding difference, but also identify the allele-specific regulatory activity. Although both technologies, whether used alone or combined, are powerful enough to screen hundreds to thousands of SNPs simultaneously, they are not able to recognize which protein or TF causes the allelic difference. To overcome this limitation, we may replace nuclear extract with a candidate protein or TF during SNPs-seq. We may also identify the candidate protein or TF through mass spectrometry-based proteomics analysis of allelically enriched protein complex.

In summary, we reported a high-throughput sequencing technology SNPs-seq for a large-scale screening of candidate SNPs to detect their allele-specific protein-binding difference. We also tested a unique application of STARR-seq to examine SNP-dependent transcriptional regulation at candidate SNP regions. The SNPs-seq along with STARR-seq provides a high-throughput pipeline for experimentally characterizing potential causal SNPs at GWAS-defined common disease loci. Knowledge gained from these technologies will facilitate translational studies for better preventive strategies and personalized clinical intervention.

## Methods

**SNP selection**. To select candidate functional SNPs at prostate cancer-risk loci, we systematically examine 146 risk SNPs and their LD SNPs ($r^2 \geq 0.5$) for a total of 6324 SNPs[22]. We first excluded any SNPs with eQTL $P$ value $\geq 1.96E−07$. We then examined these eQTL SNPs for potential overlap with prostate-specific ChIP-seq signals. The ChIP-seq data were previously collected[23], including TFs of FOXA1, AR, CTCF, ETS, EZH2, GR, JUND, NKX3_1, NR3C1, RUNX2, TCF7L2, and epigenomic marks of H3Ac, H3K4me2, H3K4me3, H3K27ac, and H4K5ac. Based on number of colocalization with ChIP-seq signals, we assigned a ChIP-seq score for each SNP and selected the candidate SNPs if ChIP-seq score was $\geq 1$, meaning at least one overlap between ChIP-seq signal and a SNP. We also examined HaploReg database (http://archive.broadinstitute.org/mammals/haploreg/haploreg.php) for additional predicted regulatory signals to prioritize the SNP selection.

**Nuclear protein preparation**. We cultured human prostate cancer cell line LNCaP in RPMI-1640 supplemented with 10% FBS and 1% penicillin/streptomycin (Life Technologies, Grand Island, NY, USA). Before androgen treatment the culture medium was replaced with phenol red free RPMI supplemented by 10% charcoal-dextran-treated FBS (HyClone, Logan, Utah, USA) and 1% penicillin/streptomycin for 3 days. Final concentration of androgen in the form of DHT (Steraloids, Newport, RI, USA) was 10 nM for treatment cells and 0.1% ethanol (ETH) for control cells. After 24 h treatment, we extracted the nuclear proteins using Ne-Per nuclear and cytoplasmic extraction reagents (Pierce Biotechnology, Rockford, IL, USA). The protein concentrations were determined using BCA protein assay kit (Pierce Biotechnology). Aliquots at 25 µl each were stored at −80 °C until use.

**Double-stranded oligo (ds-oligo) preparation**. For each selected SNP site, we synthesized four single-stranded oligos (Integrated DNA Technologies. Coralville, IA, USA) with each allele having two complementary oligos. For each oligo (21 nt), SNP site was in the middle (11th nucleotide). Concentration of each oligo was 20 µM in 25 µl duplex buffer. To make ds-oligos, we mixed 5 µl of each forward and reverse strand. After denaturation for 3 min at 95 °C, the oligo mix was subjected to annealing by gradually reducing temperature from 95 to 25 °C in 70 min. The ds-oligos were then combined in equal molar concentration to generate a large oligo pool containing all selected SNP sequences.

**Protein-bound ds-oligo isolation**. To facilitate DNA–protein binding, we mixed the ds-oligo pool with 10 µg nuclear extract in 1× incubation buffer (Affymetrix, Santa Clara, CA, USA or Signosis, Santa Clara, CA, USA) at 15 °C for 30 min before transferred to the center of the Spin Columns. After extensive washing (×6 times), the protein-bound ds-oligos were eluted in 1× Column Elution Buffer and further purified using Oligo Clean & Concentrator (Zymo Research, Irvine, CA, U.S.A.). Final concentrations of purified oligos were quantified using Qubit dsDNA HS Assay Kits (Life Technologies). We ran each assay in duplicate to test reproducibility.

**SNPs-seq library preparation and data analysis**. 2 ng of purified ds-oligo pool were subjected to sequencing library preparation using ThruPLEX DNA-seq kit (15

cycle amplification, Rubicon Genomics, Ann Arbor, MI, USA). The sequencing libraries were purified by Agencourt AMPure XP Beads (Beckman Coulter Life Sciences, Indianapolis, IN, USA) and quantified by Qubit. A pool of indexed sequencing libraries was sequenced in an Illumina HiSeq 2500 sequencer with 50 bp single read. To count sequence reads, we mapped sequencing data (fastq files) directly to 755 allele-specific sequence templates (Lasergene Genomics Suite, DNASTAR, Madison, WI, USA). Only 100% match to each oligo was allowed during the mapping. Because all sequences are unique, the perfect match will ensure complete separation of 755 oligo sequences, even for oligos with only one nucleotide difference (two different alleles for a specific SNP). We counted sequence reads for variant and reference alleles separately. To determine allele-specific binding difference, we developed a BAB score using following formula: log2 [test(RC$_{variant}$/RC$_{reference}$)/input(RC$_{variant}$/RC$_{reference}$)]. The test and input represented tested samples and input control samples, respectively. RC$_{variant}$/RC$_{reference}$ represents ratio between read count from variant allele and read count from reference allele. We defined absolute BAB score ≥ 0.58 as significant cutoff for allelic binding difference, which represents 1.50-fold difference between variant and reference alleles.

**STARR-seq plasmid library preparation**. Human STARR-seq vector was kindly provided by Dr. Stark (Research Institute of Molecular Pathology, Vienna, Austria). The plasmid vector was linearized by AgeI-HF and SalI-HF digestion (New England Biolabs, Ipswich, MA, USA), followed by agarose gel purification (QIAquick Gel extraction, Qiagen, Germantown, MD, USA) and then 1.0 × AMPure XP DNA bead purification. To prepare inserts (candidate SNP-containing sequences), we performed PCR to amplify DNA fragments covering all SNPs selected from SNPs-seq. The template was a mixed DNA from 111 prostate cancer patients to generate DNA fragments with heterozygous alleles. To facilitate cloning of the pooled PCR products, the PCR primers were designed to have an additional 15 nt recombination arms at 5′ end of each forward primer (TAGAGCATGCACCGG) and reverse primer (GGCCGAATTCGTCGA). We used highly efficient recombination-based cloning (In-Fusion HD Cloning Plus, Clontech, Mountain View, CA, USA) to construct the expression vector by mixing 50 ng pooled SNP-containing amplicons with the STARR-seq vector (100 ng). To avoid biases during the cloning, we performed a total of 8 separate recombination reactions and pooled every 4 reactions. After further purification (Agencourt AMPure XP DNA beads), we used the SNP-containing vectors (2.5 μl) to transform MegaX DH10B T1 electrocompetent bacteria (20 μl, Life Technologies). The electroporation was carried out using Gene Pulser II Electroporation system (2.0 KV, 25 μF, 200 Ω, Bio-Rad, Hercules, CA, USA). Again, we performed 8 separate transformation reactions and pooled every 4 transformations. The transformed bacteria were grown in two 500 ml LB$_{AMP}$ media overnight before extraction of the plasmid libraries that contained a pool of target sequences (Plasmid Plus Mega kit, Qiagen).

**Cell transfection for STARR-seq**. We transfected per $8.0 \times 10^6$ LNCaP cells using 20 μg plasmid DNA library and 40 μl Lipofectamine 3000 (Life Technologies). 24 h after transfection, we extracted total RNA from $3.2 \times 10^7$ cells with on-column DNase treatment (RNeasy mini kit, Qiagen). We also extracted the plasmid DNA (as input control) from $1.6 \times 10^7$ cells using Qiagen plasmid plus midi kit. By further 10-unit DNase treatment using TURBO DNase (Life Technologies) and RNA purification using RNeasy MinElute clean up kit (Qiagen), we isolated the polyA + mRNA using Ambion Dynabeads Oligo-dT25 (Life Technologies) from the total RNA. We ran each assay with technical repeat to estimate its reproducibility.

**STARR-seq library preparation and data analysis**. We used 150 ng polyA+ mRNA and performed first strand cDNA synthesis (Superscript III, Life Technologies) with a reporter RNA specific primer (GTCCAAACTCATCAATGTATC) in 16 separate reactions. After pooling every 4 reactions, we split each of the pooled cDNAs into five separate aliquots for a total of 20 PCR reactions (15 cycles, Q5 High-Fidelity DNA Polymerase, New England Biolab, Ipswich, MA, USA) using 2 reporter-specific primers (TGCTGGGATTACACATGGCAT and CTTAT-CATGTCTGCTCGAAGC) (Supplementary Fig. 10). We pooled every 5 PCR reactions and purified them using 1.0 × AMPure XP DNA beads. We sheared the PCR products into 100–150 bp by sonication and used 2 ng fragmented DNA for sequencing library preparation (15 cycles, ThruPLEX DNA-seq kit). As an input control, we amplified control plasmid DNAs isolated from transfected cells in 10 independent PCR reactions (15 cycles, 2 ng plasmid DNA per reaction) and prepared the sequencing libraries as described above. Finally, we used 1.0 × AMPure XP DNA beads to purify a total of 6 sequencing libraries including 2 DHT treatment samples, 2 ETH control samples and 2 plasmid input controls. The final libraries were sequenced on an Illumina Sequencer (HiSeq 2500) for 100 bp PE read. Sequence mapping and SNP read counting were the same as SNPs-seq. The BAE score calculation was the same as BAB score. The BAE score represents degree of transcriptional regulatory activity differences between variant and reference alleles.

**eQTL association analysis in additional datasets**. To validate the selected candidate SNPs, we performed additional eQTL analysis in three independent

prostate datasets including The Cancer Genome Atlas (TCGA), Camcap and Stockholm cohorts which comprised of 389, 119, and 94 prostate samples, respectively[23]. We used Matrix eQTL to test the cis-eQTL associations and parameters "useModel = modelLINEAR", "errorCovariance = numeric ()" were applied[45]. In addition, we applied the non-parametric Kruskal–Wallis H test to assess the statistical significance between the gene expression and SNP genotypes. For haplotype (rs6579003, rs7123299, rs7123418) analysis, we first defined haplotypes for each patient using MACH1[46] and minimac program[47,48] in 1000 Genomes Project Phase I V3 EUR reference ($n = 379$) and then performed linear regression analysis, regressing normalized expression levels on the number of minor alleles of each SNP/haplotype genotype. R (version 3.2.2) was used to perform the statistical tests and graphically visualize the association between SNP genotypes and gene expression levels. The RGS17 and ASCL2 mRNA levels were assessed by RNA-seq in TCGA, Illumina Expression BeadChip-based transcriptional profiling in Camcap and Stockholm cohorts of human prostate tissue samples.

**Cell culture for functional analysis**. The LNCaP, 22Rv1, and VCaP cells were originally obtained from the American Type Culture Collection (ATCC, Manassas, VA, USA) and confirmed to be mycoplasma-free during the experiments. The cell culture condition was at 37 °C with 5% CO$_2$. Specifically, LNCaP and 22Rv1 cells were grown in RPMI-1640 (Sigma-Aldrich, St. Louis, MO, USA), VCaP cells were grown in DMEM (Invitrogen, Carlsbad, CA, USA). Ten percent FBS and antibiotics (penicillin and streptomycin, Sigma-Aldrich) were added to the base media. To study AR activity and stimulate androgen signaling in these cell lines, we cultured cells in charcoal-stripped medium for up to 48 h, then the cells were treated with 100 nM DHT (dissolved in ethanol).

**Luciferase reporter assay**. To validate allele-dependent regulatory differences in selected candidate SNPs, we applied dual-luciferase reporter assay system by cloning SNP-containing sequences into vector pGL4.28 (Promega, Madison, WI, USA). We used the pGL4.74 as an internal control and measured the luciferase activity of the transfected cells according to the manufacturer's protocol on a bioluminometer. We also replaced minimal promoter in pGL4.28 with target gene RGS17 promoter (chr6:153131052–153132096, hg38) to evaluate direct effect of a candidate SNP on its target gene. All reading measurements were obtained from at least three replicates. Statistical analysis of significance was determined by two-tailed student's t-test using IBM SPSS statistics software version 24.

**Chromatin immunoprecipitation (ChIP)**. To test for TF-binding status at selected SNP sites, we fixed the LNCaP and VCaP cells using 1% formaldehyde for 10 min and stopped the fixation using 125 mM glycine at room temperature. To isolate nuclei, we suspended the cell pellet in hypotonic lysis buffer (Sigma-Aldrich) for 45 min. The nuclei were washed twice by cold PBS and then suspended in SDS lysis buffer. The nuclei chromatin was sonicated to an average of 400 bp. To make dynabeads–antibody complex, we first washed dynabeads (Dynabeads Protein A/G for Immunoprecipitation, Invitrogen) twice by blocking buffer, and then incubated the beads with 7 μg antibodies for 10 h at 4 °C. 250 μg of sonicated chromatin was diluted in IP buffer to final volume of 1.35 ml, then added to 70 μl of Dynabeads–antibody complex (~1:400 dilution for the antibodies with 2 μg/μl concentration in stock). After 12 h incubation at 4 °C, the complex was washed once with wash buffer I and buffer II, followed by two more washings with buffer III and buffer IV. The DNA–protein complex was separated from beads by extraction buffer, then DNA and protein were reverse cross-link with Proteinase K and NaCl overnight at 65 °C. The DNA was purified by MinElute PCR Purification Kit (Qiagen). The buffers and antibodies are listed in Supplementary Tables 1 and 2.

**ChIP allele-specific quantitative PCR (ChIP-AS-qPCR)**. To confirm TF-binding at selected SNP sites, we performed ChIP-AS-qPCRs to quantify TF-binding differences between variant and reference alleles. We designed PCR primers to amplify the DNA fragments harboring the different alleles at these SNP sites. We performed the quantitative PCRs at each SNP site in triplicates. We determined the relative enrichment of each candidate TF at target DNA fragments by comparing to IgG controls. The qPCR and allele-specific qPCR primers are listed in Supplementary Table 3.

**CRISPR interference (CRISPRi) at rs13215402 site**. To test direct effect of a candidate SNP on RGS17 mRNA expression, we applied online tool (http://crispr.mit.edu/) and designed small guide RNAs (sgRNAs) targeting 20 bp at SNP site of interest[49]. We cloned the sgRNAs into the pLV hU6-sgRNA hUbC-dCas9-KRAB-T2a-Puro plasmid (Addgene, 71236). To confirm genotype of individual clones, we performed sequencing analysis with hU6 primer. The constructed plasmids were used to transfect LNCaP cell lines with Lipofectamine 3000 (Thermo Fisher Scientific, Waltham, MA USA). 48 h after transfection, we isolated total RNA and performed RT-qPCR using SuperScript VILO cDNA Synthesis Kit (Thermo Fisher Scientific). We performed each transfection in triplicates and each RT-qPCR in triplicates. We determined relative gene expression using the ΔΔCt method with ACTB as a control. sgRNA and PCR primer sequences are listed in Supplementary Table 4.

**Single nucleotide editing at rs13215402 site**. To further examine direct effect of a candidate SNP on *RGS17* expression, we inserted sgRNAs into pSpCas9(BB)-2A-Puro (PX459) V2.0 (Addgene, 62988)[50]. The single-stranded oligodeoxynucleotides (ssODNs) that were centered at rs13215402 (either A or G) were used as repair templates (Supplementary Table 4). We co-transfected 70% confluence of prostate cancer cells 22Rv1 with the 300 ng Cas9 plasmid (sgRNA rs13215402 A or G) and 1 μl of ssODN template (10 μM) using Lipofectamine 3000. The medium was changed after overnight incubation followed by adding 0.8 μg per ml puromycin (Sigma) into the transfected cells. 48 h after puromycin treatment, we seeded the single cells in 96-well plates and checked single clonality to exclude non-single clones within 9–16 days. Eventually, we selected the single clones for subculture and performed Sanger sequencing for genotype examination within 1 to 2 months. To quantify *RGS17* expression after genotype changes, we performed RT-qPCR in selected subclones with different genotypes.

**siRNA transfection and cell proliferation assays**. To evaluate functional consequence of selected target gene, we performed siRNA-mediated knockdown assay in prostate cancer cell lines and determined the effect of the gene knockdown on cancer cell proliferation. 24 h before transfection, we seeded 50–60% confluent LNCaP or 22RV1 cells. We reversely transfected LNCaP or 22RV1 cells ($2.5 \times 10^3$ per well) with control, cell death, *RGS17* siRNAs using HiPerFect transfection reagent (Qiagen). We changed the medium after 24 h and collected the cells after 48 h to test siRNA-mediated knockdown efficiency. To determine the cell viability and proliferation, we applied XTT (Roche Diagnostics GmbH, Mannheim, Germany) reagent and measured the absorbance at 450 nm at a designated time point following manufacturer's instruction. Two-tailed *t*-test was used to calculate the significances. siRNAs are listed in Supplementary Table 5.

**Clinical association analysis for target gene expression**. To estimate clinical relevance of *RGS17*, we examined the association of the gene expression with prostate cancer and clinicopathological features using RNA profiling data from TCGA (from The cBio cancer genomics portal[51]). We applied the non-parametric Mann–Whitney *U* test to evaluate the significance of gene expression levels between 52 normal and 497 tumor tissues. R (version 3.2.2) was used to perform statistical analyses and box plot was used to graphically display gene expression intensities (log base 2) between different groups. In addition, we evaluated the association of target gene expression with tumor stage, Gleason score, prostate-specific antigen (PSA), and the severity in TCGA dataset. To assess the potential association between target gene expression levels and prostate cancer survival, we applied the non-parametric statistic Kaplan–Meier estimator and tested 162 stage 1 patients in TCGA data and 79 patients from the Glinsky[52] in Oncomine database[53]. Samples were stratified into two groups based on the mean values of *RGS17* expression levels. R (version 3.2.2) and R package "Survival" were used for the analysis.

**Data availability**. All relevant data are available within the article and supplementary files, or available from the authors upon request.

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

## Acknowledgements

This study was supported by the National Institutes of Health (R01 CA157881) and the Advancing a Healthier Wisconsin (Project# 5520227) awarded to L.W., the Academy of Finland (284618 and 279760) and University of Oulu Strategic Funds and Jane & Aatos Erkko Foundation awarded to G.-H.W., the National High-Tech Research and Development Program of China (SQ2015AA0202183) and the Doctoral Team Foundation of the First Affiliated Hospital of Zhengzhou University (2016-BSTDJJ-03) awarded to L.-D. W., and the China Scholarship Council (No. 201407040025) awarded to P.Z.

## Author contributions

L.W., G.-H.W., and L.-D.W. designed the experiments, and obtained financial support. P. Z., J.-H.X., J.Z., P.G., Y.-J.T., M.D., Y.-C.G., S.S., Q.Z., L.S.T., and A.J.F. performed the experiments. P.Z., J.-H.X., J.Z., P.G., Y.-J.T., M.D., Y.-C.G., S.S., Q.Z., L.S.T., A.J.F., and J. R.C. analyzed the data. P.Z., J.-H.X., J.Z., P.G., Y.-J.T., M.D., S.S., Q.Z., M.K., L.S.T., S.N. T., and A.J.F. contributed reagents/materials/analysis tools. L.W., G.-H.W., and P.Z. wrote the original draft. All authors reviewed and edited the paper.

## Additional information

**Competing interests:** The authors declare no competing interests.

