## [Peer Review File · Nature Communications]

Reviewers' comments:

Reviewer #1 (Remarks to the Author):

Authors propose a new high-throughput sequencing protocol for parallel screening of hundreds to thousands of SNPs (SNPs-seq). Strength of the method is the ability to precisely determine allelic protein-binding differences in high-throughput manner.

In paper authors also test STARR-seq for validation of allele specific expression at the regulatory (enhancer) sequences and experimentally validate one of the indicated target gene.

Overall, the paper would benefit of more clear focus on and validation of SNPs-seq, rather than also extending the analysis to STARR-seq as well (that also raises questions about additional validation). In addition, to be exact the experimental validation of RGS17 validates the effect of gene expression alteration, not the alteration of the SNP.

Specific comments:

Direct comparison to other available techniques that address allele specific binding of TFs is lacking (e.g. to ChIP-exo / ChIP-nexus data sets). Are the specificity and sensitivity similar? While in principle the proposed approach is much more general than analysis of single factor by ChIP, this is not clearly demonstrated in the manuscript. This is highlighted by the fact that authors choose to experimentally validate two loci that are clearly indicated by the binding of well known prostate cancer associated TFs (AR, FOXA1, HOXB13).

While authors demonstrate good correlation between replicates, reasons for variation in read counts is not addressed sufficiently. E.g. authors state: "Median read count per allele was 4,818, ranging from 0 to 108,181 in 20 different SNPs-seq assays" This variation and especially issue with zero read counts should be discussed in more details. What are the sources of bias that may results in zero reads? Authors do list the sources of variation, but detailed analysis to isolate the effect of variation is lacking; Only GC bias is discussed. This is highly relevant for the applicability of the method.

I would also like to see data on how the method performs in terms of the quality metrics when the number of assayed SNPs is altered. Authors show data with 755 unique sequences (374 SNPs); scaling to thousands of SNPs is not demonstrated.

Reviewer #2 (Remarks to the Author):

This manuscript proposed an innovative method named SNPs-seq to identify variants that cause transcription factor binding difference. When further coupled with STARR-seq, the integrated pipeline is also claimed to detect variant-dependent transcription regulation. Methods like SNPs-seq are likely to play an important role in genomic variant analysis and precision medicine, such as to detect novel prognostic biomarkers, if the following points on the manuscript could be further clarified:

1. The compilation process needs more definition and explanation on why certain SNPs were chosen. For example, how exactly were the ChIP-seq data and HaploReg database used to identify potential SNPs.
2. The type of SNPs that can undergo screening by SNPs-seq. The manuscript needs to clarify if SNPs from exons, introns and 3'UTR regions can also be screened by SNPs-seq. And since only

prostate cancer loci were tested, will this method work in other physiological conditions?

3. The reads mapping process. It seems the oligo length is quite short. The readers need more to know how the raw reads were aligned. How is multi-mapping handled? Reads with multi-mismatch?

4. Time and Cost. The average turnaround time and cost for SNPs-seq.

5. Comparison with existing or similar method. The manuscript mentions "Although these current methods have enabled functional characterization of regulatory variants at some GWAS loci, the progress is extremely slow." The reads will benefit from a more thorough review of existing methods.

Reviewer #1 (Remarks to the Author):

Authors propose a new high-throughput sequencing protocol for parallel screening of hundreds to thousands of SNPs (SNPs-seq). Strength of the method is the ability to precisely determine allelic protein-binding differences in high-throughput manner. In paper authors also test STARR-seq for validation of allele specific expression at the regulatory (enhancer) sequences and experimentally validate one of the indicated target gene.

1. Overall, the paper would benefit of more clear focus on and validation of SNPs-seq, rather than also extending the analysis to STARR-seq as well (that also raises questions about additional validation).

Response: To focus on SNPs-seq, we have moved all STARR-seq related methods and figures to Supplementary materials. We have moved some Supplementary Figures to main text to strengthen the SNPs-seq. We also regrouped some Figures to be more logically displayed.

2. In addition, to be exact the experimental validation of RGS17 validates the effect of gene expression alteration, not the alteration of the SNP.

Response: To address this question, we performed a series of tests and evaluated direct effect of rs13215402 on RGS17 expression. First, we tested if the putative SNP-containing enhancer has a direct effect on RGS17 promoter itself instead of a universal minimal promoter used in initial test (Fig. 4c). We replaced the minimal promoter in pGL4.28 with the RGS17 promoter and tested the SNP effect on the target gene promoter directly. This test showed significantly higher relative luciferase signal in allele G than allele A, regardless of DHT treatment (Fig. 4d). This result was consistent with initial result that used a universal minimal promoter (Fig. 4c).

Second, we applied CRISPR interference (CRISPRi) technology to evaluate the repression effect by interfering the SNP region on the RGS17 expression. We constructed plasmids (CRISPR/dCas9) containing sgRNAs that targeted the SNP sequence and transfected LNCaP cell line (heterozygous for rs13215402). We then quantified the expression of RGS17 and observed its downregulation by either allele G or allele A interference. When compared to non-target controls (NTC), the downregulation was significant when targeting allele G ($p=0.028$) but not allele A ($p=0.633$) (Fig. 4e). To further delineate the regulatory role of the SNP, we applied CRISPR/Cas9-based genome editing technology in the prostate cancer cell line 22Rv1 (also heterozygous for rs13215402) with an aim of creating subclones with homozygous alleles (A/A or G/G) and determining direct effect of these different genotypes on RGS17 expression. During the last three months, we were able to generate two subclones with genotype G/G but did not obtain subclones with genotype A/A. The quantitative RT-PCR analysis showed 3-fold increase of RGS17 mRNA levels in subclones with homozygous G/G when compared to parental cell line with heterozygous G/A (Fig. 4f). These new data strongly support the regulatory role of rs13215402 on RGS17 expression. We have added these experimental procedures and results in revised manuscript on pages 13, 25, 26 and 27.

Fig. 4. Allele-dependent transcriptional regulation at rs13215402.

Specific comments:

3. Direct comparison to other available techniques that address allele specific binding of TFs is lacking (e.g. to ChIP-exo / ChIP-nexus data sets). Are the specificity and sensitivity similar?

Response: It is difficult to directly compare SNPs-seq with ChIP-seq data. SNPs-seq in the current study detects a combined effect of unknown nuclear proteins (one or multiple proteins/TFs) while ChIP-seq assays detect allelic binding difference of one specific TF. If two competing TFs have an opposite allelic binding preference, SNPs-seq may show allele preference toward the protein with stronger binding capacity or may even miss the detection of allelic binding preference. Since SNPs-seq (if using nuclear extract) will not tell which TF (or

TFs) binds to a selected SNP allele, direct comparison between SNPs-seq (unknown TFs) and ChIP-seq (known TF) is not possible.

4. While in principle the proposed approach is much more general than analysis of single factor by ChIP, this is not clearly demonstrated in the manuscript. This is highlighted by the fact that authors choose to experimentally validate two loci that are clearly indicated by the binding of well known prostate cancer associated TFs (AR, FOXA1, HOXB13).

Response: *We agree with the comment that SNPs-seq is a general screening tool. We have discussed this limitation and provided potential resolutions on page 19.*

Page 19: "...To overcome this limitation, we may replace nuclear extract with a candidate protein or TF during SNPs-seq. We may also identify the candidate protein or TF through mass spectrometry-based proteomics analysis of allelically enriched protein complex."

5. While authors demonstrate good correlation between replicates, reasons for variation in read counts is not addressed sufficiently. E.g. authors state: "Median read count per allele was 4,818, ranging from 0 to 108,181 in 20 different SNPs-seq assays" This variation and especially issue with zero read counts should be discussed in more details. What are the sources of bias that may result in zero reads? Authors do list the sources of variation, but detailed analysis to isolate the effect of variation is lacking; Only GC bias is discussed. This is highly relevant for the applicability of the method.

Response: *Sources of bias that may result in low read counts include multiple factors. In addition to GC bias, we further analyzed 5'/3' nucleotide composition and thermodynamic features of each oligo. This analysis demonstrated significant association of read counts with thermodynamic parameters of these oligos including Tm, and delta G (thermostability). This analysis also showed that the oligos with A, T, AA, AT, TA, or TT at their 5' and/or 3' ends had significant lower read counts. These results indicated overall GC content (GC%, Tm, and thermostability) and 5'/3' end composition have a significant effect on read count. We have added these results on page 10 of the revised manuscript including **Fig. 3 (e, f)** and **Supplementary Fig. 3 (c, d)**.*

*Page 10: "...Accordingly, the read counts were also associated with Tm values ($R^2=0.6458$, **Supplementary Fig. 3c**). We also calculated delta G using online program (<http://primerdigital.com/tools/>)²⁴ to determine thermostability of each oligo and observed a significant association ($R^2=0.6526$, **Fig. 3e**). Clearly, higher thermostability of these single stranded oligos increased annealing efficiency to form double stranded oligos, hence providing more templates for sequencing library preparation.*

*We then tested the effect of nucleotide compositions at 5'/3' ends on read counts. We grouped the 755 oligos into different groups, based on their nucleotide compositions. When compared to G/C, single nucleotide A/T at either 5' or 3' end significantly reduced read counts of corresponding oligos. When comparing 16 possible two nucleotide combinations at either 5' or 3' ends, the combinations of AA, AT, TA and TT showed the lowest read counts (**Fig. 3f**, **Supplementary Fig.3d**). Clearly, overall GC content (GC%, Tm, and thermostability) and 5'/3' end composition have a significant effect on read count in the final sequencing libraries."*

Fig. 3 SNPs-seq data analysis and BAB score distribution

6. I would also like to see data on how the method performs in terms of the quality metrics when the number of assayed SNPs is altered. Authors show data with 755 unique sequences (374 SNPs); scaling to thousands of SNPs is not demonstrated.

Response: In fact, we have also applied the SNPs-seq in a separate study. By working with our collaborator (Dr. Stephen Thibodeau at Mayo Clinic), we have applied the SNPs-seq to screen 1,274 SNPs (2,548 ds-oligos) that were selected from six prostate cancer risk loci. In this study, all SNPs (minor allele frequency ≥ 0.05) in the six risk regions were chosen for SNPs-seq analysis. Consistent with the results from 374 SNPs, the test in 1,274 SNPs demonstrated high correlation between technical replicates (See following fig. a). Importantly, 48 SNPs were tested in both 374 SNP set and 1,274 SNP set. Read counts from the 48 overlapped SNPs (total 96 alleles) showed a significant correlation in their allele-based read counts (See following fig. b). This result suggests that SNPs-seq can test at least over a thousand of SNPs in a single assay.

Since 1,274 SNP data set is for a separate study, it is not possible to include this data in this revised manuscript. However, we will publish this data set in a separate manuscript.

Reviewer #2 (Remarks to the Author):

This manuscript proposed an innovative method named SNPs-seq to identify variants that cause transcription factor binding difference. When further coupled with STARR-seq, the integrated pipeline is also claimed to detect variant-dependent transcription regulation. Methods like SNPs-seq are likely to play an important role in genomic variant analysis and precision medicine, such as to detect novel prognostic biomarkers, if the following points on the manuscript could be further clarified:

1. The compilation process needs more definition and explanation on why certain SNPs were chosen. For example, how exactly were the ChIP-seq data and HaploReg database used to identify potential SNPs.

Response: We have revised the description of SNP selection to show the selection process more clearly. Specifically, we extensively modified the selection process in Method section on page 21: “To select candidate functional SNPs at prostate cancer risk loci, we systematically examine 146 risk SNPs and their LD SNPs ($r^2 > 0.5$) for a total of 6,324 SNPs. We first excluded any SNPs with eQTL P value $\geq 1.96E-7$. We then examined these eQTL SNPs for potential overlap with prostate-specific ChIP-seq signals. The ChIP-seq data were previously collected (Whittington T, et al. Nat Genet 48, 387-397 (2016)), including TFs of FOXA1, AR, CTCF, ETS, EZH2, GR, JUND, NKX3_1, NR3C1, RUNX2, TCF7L2 and epigenomic marks of H3Ac, H3K4me2, H3K4me3, H3K27ac, and H4K5ac. Based on number of colocalization with ChIP-seq signals, we assigned a ChIP-seq score for each SNP and selected the candidate SNPs if ChIP-seq score was ≥ 1 , meaning at least one overlap between ChIP-seq signal and the SNP. We also examined HaploReg database (<http://archive.broadinstitute.org/mammals/haploreg/haploreg.php>) for additional predicted regulatory signals to prioritize the SNP selection.”

We also updated result section accordingly on page 8.

2. The type of SNPs that can undergo screening by SNPs-seq. The manuscript needs to clarify if SNPs from exons, introns and 3'UTR regions can also be screened by SNPs-seq. And since only prostate cancer loci were tested, will this method work in other physiological conditions?

Response: *The selected candidate SNPs in this study included 128 at intergenic, 12 at upstream of gene, 5 at 5'-UTR-exon, 4 at coding-exon, 213 at intron, 9 at 3'-UTR-exon, and 3 at downstream of gene. This information has been included in **Supplementary Table 1**.*

We believe that SNPs-seq is also applicable to other physiological conditions. The reason is that specific binding of a protein (TF) to a unique dsDNA is a universal mechanism of gene regulation. Due to cell/tissue type-specific nature of gene regulation, however, SNPs-seq may detect one set of SNPs with allelic protein binding in one physiological condition but another set of SNPs in different condition. In other words, the SNPs-seq will work in any physiological conditions but SNPs detected by SNPs-seq will be physiological conditions-dependent.

We have added the following sentences in discussion section to further strengthen its applications (page 18): "...It is worth mentioning that regulation of gene expression is cell and tissue type-specific, so is allelic binding preference detected by SNPs-seq. When applying the SNPs-seq to detect allele-specific binding difference, cell/tissue type and surrounding environments that are involved in the diseases/phenotypes of interest should be taken into consideration."

*In addition, to test the SNPs-seq assay in other physiological conditions, another ongoing study in our lab has tested 83 candidate SNPs at lung cancer risk loci. We performed this analysis using nuclear extract from a lung cancer cell line (A549). This result showed highly correlation ($R^2 > 0.99$) between technical replicates, either in test samples (1 vs 2) (**Following Figure a**) or input controls (1 vs 2) (**Following Figure b**). Since it is for a lung cancer study, we will not include this result in the revised manuscript.*

3. The reads mapping process. It seems the oligo length is quite short. The readers need more to know how the raw reads were aligned. How is multi-mapping handled? Reads with multi-mismatch?

Response: We applied the Lasergene Genomics Suite (DNASTAR) for mapping and read counting. The template was 755 unique oligo sequences (21nt) in fasta format. We called a mapped read if a sequence has 100% match to one of the 755 sequences (not whole genome sequence). The perfect match will ensure complete separation of oligo sequences with only one nucleotide difference. Since all oligos are unique and only perfect match is allowed, there is no multi-mapping or multi-mismatch issues. We have revised our mapping description to clarify this read calling process on page 23.

4. Time and Cost.

Response: We have added the time and cost estimation as a **Supplementary Table 5**.

Time estimation after candidate SNPs are selected:

Week 1: oligo synthesis, cell culture and nuclear extract preparation

Week 2-3: oligo preparation, oligo-protein binding and sequencing library preparation

Week 4: sequencing and data analysis

Cost estimation for 374 SNPs in 20 assays (excluding labor)

Company	Cat.No.	Reagents	Q	Price(USD)	Total items	Cost per 20 tests
Signosis	A-1002	A-1002, Transcription Factor Activation Profiling columns and column buffers: 20 columns, filter wash buffer, filter binding buffer, and elution buffer.	1	\$350.00	20 columns	\$350.00
IDT	NA	4 plate oligos (384 well plate, 1,510 oligo, 21nt length)	1	\$2,581.80	25ul per oligo	\$2,581.80
Pierce Biotechnology	78833	NE-PER Nuclear and Cytoplasmic Extraction Reagents	1	\$219.00	15,000 ul	\$87.60
Zymo Research	D4060	Oligo Clean & Concentrator (50 Preps) w/ Zymo-Spin IC Columns	1	\$97.00	50 columns	\$38.80

Life Technologies	Q32851	Qubit dsDNA HS Assay Kits	1	\$89.00	250 ul	\$21.36	
Rubicon Genomics	R400427	ThruPLEX DNA-seq 48S Kit	1	\$1,920.00	48 Rxns	\$800.00	
Beckman Coulter Life Sciences	A63881	Agencourt AMPure XP	1	\$1,190.00	60,000 ul	\$29.75	
Illumina	HiSeq2500 sequencer	50bp single read kit	1	\$1,600.00	48	\$1,600.00	
						Total cost	\$5,509.31
						Cost per SNP	\$14.73

5. Comparison with existing or similar method. The manuscript mentions “Although these current methods have enabled functional characterization of regulatory variants at some GWAS loci, the progress is extremely slow.” The readers will benefit from a more thorough review of existing methods.

Response: *The existing methods include low throughput assays (EMSA, reporter assay, CRISPR/Cas9) and high throughput assays (ChIP-seq, DNase-seq and their derivatives such as ChIP-exo and ChIP-nexus). In addition to the descriptions of each method, we have added the following paragraph in introduction and discussion sections.*

Page 5 in Introduction: “One existing high-throughput method is to calculate allele-specific read counts from available ChIP-seq data. Significant deviation of read counts between two alleles indicates allelic binding preference for the unique TF. However, to be informative, the SNPs of interest need to be heterozygous in the tested cell line. For a large group of SNPs, it is difficult to find cell lines with heterozygous status in all (or most) candidate SNPs.” ...

Page 17-18 in Discussion: “...Although useful for mapping regulatory genomic regions, none of these datasets provide direct access to allelic binding preferences. One potential solution is to extract read counts at SNP sites from these sequencing data. To date, ENCODE project has collected over six thousands of epigenomic profiling data including ChIP-seq and DNase-seq. Systematic examination of these sequencing data may reveal allele-specific transcriptional control at each SNP site. However, for a specific disease type, there may be no or few epigenomic profiling data available. Importantly, most current epigenomic profiling datasets are based on a handful of cell lines. If a cell line used is not heterozygous for a specific SNP, it will not generate any allelic information. For example, only 96 (25.67%) of 374 candidate SNPs selected in this study are heterozygous in LNCaP cell line. Furthermore, most current epigenomic profiling datasets have relative low coverage at a certain SNP sites, which makes determination of allele-specific binding difficult.”

REVIEWERS' COMMENTS:

Reviewer #1 (Remarks to the Author):

Authors have successfully addressed all my concerns with their revision.

Reviewer #2 (Remarks to the Author):

The authors have addressed all my concerns.